# The Application of Systems Thinking to the Prevention and Control of Sexually Transmissible Infections among Adolescents and Adults: A Scoping Review

**DOI:** 10.3390/ijerph20095708

**Published:** 2023-05-02

**Authors:** Daniel Vujcich, Meagan Roberts, Tyler Selway, Barbara Nattabi

**Affiliations:** 1Western Australian Sexual Health and Blood-Borne Virus Applied Research and Evaluation Network, School of Population Health, Curtin University, Perth, WA 6102, Australia; meagan.roberts@curtin.edu.au (M.R.);; 2School of Population and Global Health, University of Western Australia, Perth, WA 6009, Australia; barbara.nattabi@uwa.edu.au

**Keywords:** HIV, sexually transmissible infections, sexual health, systems thinking, review

## Abstract

Systems thinking is a mechanism for making sense of complex systems that challenge linear explanations of cause-and-effect. While the prevention and control of sexually transmissible infections (STIs) has been identified as an area that may benefit from systems-level analyses, no review on the subject currently exists. The aim of this study is to conduct a scoping review to identify literature in which systems thinking has been applied to the prevention and control of STIs among adolescent and adult populations. Joanna Briggs Institute guidelines for the conduct of scoping reviews were followed. Five databases were searched for English-language studies published after 2011. A total of *n* = 6102 studies were screened against inclusion criteria and *n* = 70 were included in the review. The majority of studies (*n* = 34) were conducted in African nations. Few studies focused on priority sub-populations, and 93% were focused on HIV (*n* = 65). The most commonly applied systems thinking method was system dynamics modelling (*n* = 28). The review highlights areas for future research, including the need for more STI systems thinking studies focused on: (1) migrant and Indigenous populations; (2) conditions such as syphilis; and (3) innovations such as pre-exposure prophylaxis and at-home testing for HIV. The need for conceptual clarity around ‘systems thinking’ is also highlighted.

## 1. Introduction

Systems thinking is a mechanism for making sense of complex systems that challenge linear or reductionist explanations of cause-and-effect [1]. System complexity may be related to several factors, including contextual dynamism, the presence of multiple, interconnected agents, or the capacity of elements within the system to adapt and change over time [1,2]. A bibliometric analysis reveals that systems thinking has been most applied in the fields of business and management [1]. While repeated calls have been made to apply systems thinking to public health issues, two systematic reviews have found relatively few examples of its actual application in that context [3,4]. A review by Carey and colleagues identified 117 articles in which systems thinking was applied to public health, but almost half of these were commentary pieces or calls for greater use of the concept [3]. Similarly, Rusoja and colleagues’ review of the application of systems thinking in the academic health literature identified 166 articles with a public health focus but nearly all were theoretical discussions of systems thinking [4].

It is possible that relevant publications have been omitted from public health literature reviews relating to systems thinking due to the search strategies adopted. For instance, in the systematic review conducted by Carey and colleagues, the search strategy combined systems thinking concepts with “public health and key domains of public health activity (i.e., obesity, tobacco, alcohol, and social determinants of health)” [3] (p. 2). Consequently, the search strategy may not have been sufficiently sensitive to public health issues unrelated (or indirectly related) to those domains. As Carey and colleagues note, no review can provide an exhaustive list of the literature on a subject; instead, all reviews are necessarily ‘bounded’ by factors including search strategy design [3].

While only one of the studies included in Carey and colleagues’ review of the public health literature specifically related to sexually transmissible infections (STIs) [5], the prevention and control of STIs has been identified as an area that may benefit from systems-level analyses [6]. STI epidemiology has been shown to be influenced by dynamic sexual networks and patterns of sexual mixing which are, in turn, influenced by changing socio-cultural contexts, and the introduction of new technologies [6]. Moreover, reductions in the transmission of one STI can coincide with an increase in the transmission of another, as seen in the rising rates of syphilis in some populations following the introduction of effective human immunodeficiency virus (HIV) medications such as highly active antiretroviral therapy (HAART) and pre-exposure prophylaxis (PrEP) [6,7].

A preliminary search of the literature finds that some relevant systems thinking studies with a focus on STIs were not included in the more general reviews of the public health literature [8,9,10,11]. Consequently, a more focused review on the application of systems thinking in the prevention and control of STIs is warranted. A search of Embase, the Cochrane Database of Systematic Reviews, and JBI Evidence Synthesis suggests that no existing review on the subject exists or is under development.

The aim of this study is to conduct a scoping review to identify literature in which systems thinking has been applied to the prevention and control of STIs among adolescent and adult populations globally. The purpose of the review is to answer the following research questions:(1)Which specific STIs have been studied from a systems thinking perspective?(2)In which populations were the studies conducted (i.e., country, age group, gender, sexuality, and ethnic/Indigenous identity)?(3)Which systems thinking concepts and/or tools have been utilized or referenced?

The overarching aim and specific research questions align with the indications for a scoping review [12]. According to Munn and colleagues, scoping reviews can be used as a precursor to a systematic review and are useful for identifying: methods used to investigate a specific topic; significant features associated with a concept; and knowledge gaps [12].

## 2. Materials and Methods

### 2.1. Search Strategy

Methodological guidelines for the conduct of scoping reviews were followed and used to develop a protocol (summarized here and available on request) [13]. Due to resource constraints, the review was focused on published and unpublished English language literature released after 1 January 2012. The lower date limit (2012) represents the year in which the STI prevention and control landscape changed considerably with the approval of HIV home self-testing kits and the use of Truvada for PrEP in the United States of America [14].

Existing reviews about either STIs or systems thinking were used to identify relevant search terms [1,3,4,15,16]. The terms were incorporated into database-specific search strategies with input from a research librarian (see Appendix A: Search strategy and results, by database). The databases Embase, MEDLINE, Web of Science Core Collection, and Google Scholar have been shown to provide adequate and efficient coverage and were searched on 22 December 2022 [17]. Only the first 200 results of Google Scholar were marked for screening based on methodological guidance [18]. Additionally, the Cochrane Library was searched to identify relevant protocols.

While Google Scholar has been shown to be useful in identifying grey literature, the search engine has limitations and supplementary approaches to grey literature searching have been recommended [18]. The main risk of excluding grey literature is that it can yield an unbalanced picture of the available evidence in so far as studies with null or negative results are less likely to be published [19]. Given that this scoping review is principally focused on identifying study characteristics (as opposed to synthesizing results), the reliance on Google Scholar was considered sufficient and practical given resource constraints.

The results yielded in database searches were exported into a reference management system and deduplicated using the process developed by Bramer and colleagues [20,21]. The deduplicated list of studies was then imported into the Joanna Briggs Institute System for the Unified Management, Assessment, and Review of Information (JBI SUMARI) for screening by two independent reviewers against inclusion and exclusion criteria [22]. An initial pilot test was conducted with a sample of 100 studies to discuss any conflicts with a view to ensuring that reviewers had a consistent understanding of the inclusion and exclusion criteria. Any screening conflicts after the pilot stage were resolved by consensus among the relevant reviewers.

### 2.2. Title and Abstract Screening

The titles and abstracts of English-language studies published/released after 1 January 2012 were screened against the following eligibility criteria.

#### 2.2.1. Participants

Studies were included if they related to populations of adolescents or adults. Consistent with the World Health Organization definition, adolescence is considered to commence at 10 years of age and is the phase during which patterns of behaviour relevant to sexual health begin to be established [23]. No geographical or other demographic limitations were applied.

#### 2.2.2. Concept

Studies were included if they involved the application of systems thinking. While there are various understandings of systems thinking, the definition proposed by Arnold and Wade was adopted on the basis that it both synthesizes key components of existing definitions and explains how the goal of systems thinking distinguishes it from other approaches:
Systems thinking is a set of synergistic analytic skills used to improve the capability of identifying and understanding systems (i.e., groups or combinations of interrelated, interdependent, or interacting elements forming collective entities), predicting their behaviours, and devising modifications to them in order to produce desired effects.[24] (p. 675)

For the purposes of this review, the skills used to identify, understand, predict the behaviour of, or devise modifications to a system are those represented in Arnold and Wade’s systemigram—namely:*Recognizing interconnections:* recognizing key connections between system parts.*Identifying and understanding feedback:* identifying cause-and-effect feedback loops and their impact on the system.*Understanding system structure:* understanding the elements of a system, how they are connected, and how they contribute to the way a system behaves.*Differentiating types of stocks, flows, and variables:* identifying pools of resources (stocks) in a system, changes in the amounts of those resources (flows), and the factors (variables) that contribute to those changes in both linear and non-linear ways.*Understanding dynamic behaviour:* understanding how feedback loops both influence and are influenced by stocks, flows, and variables.*Reducing complexity by modelling systems conceptually:* using methods such as reduction, transformation, abstraction, and homogenization to conceptually model a complex system in a more simplified and accessible manner.*Understanding systems at different scales:* recognizing that systems can be examined in different levels of detail and can be embedded within larger systems [24].

Papers were not considered for inclusion if they merely referenced ‘systems’ (e.g., health systems or socio-political systems) without explicitly adopting the aforementioned approaches to identify, understand, predict the behaviour of, or devise modifications to a system.

#### 2.2.3. Context

Only studies in which systems thinking is used in the context of efforts to prevent and control STIs were considered for inclusion. Here, STIs are defined as infections caused by “bacteria, viruses, or other microorganisms that can be passed from one person to another through blood, semen, vaginal fluids, or other body fluids, during oral, anal, or genital sex” [25]. Prevention can be primary (focused on avoiding an STI), secondary (focused on halting progress of an STI at an early stage), or tertiary (focused on minimizing STI progression and improving outcomes) [26]. From a public health perspective, the aim of prevention is disease control. For the purposes of this review, the definition of control set out in the 1997 Dahlem Workshop on the Eradication of Infectious Diseases was adopted—namely, “the reduction of disease incidence, prevalence, morbidity or mortality to a locally acceptable level as a result of deliberate efforts; [and] continued intervention measures … required to maintain the reduction” [27].

### 2.3. Full-Text Screening

The full text of studies identified for possible inclusion at the title and abstract screening stage was then screened by two independent reviewers. However, after screening approximately 40 studies, it became apparent that many did not apply the tools and methods that are typically associated with systems thinking. Consequently, it was often difficult to judge whether studies could be said to apply “systems thinking”. For instance, several studies examined individual, health service, and societal barriers to engagement in STI prevention and control initiatives and could therefore potentially be interpreted as recognizing key connections between system parts or understanding the elements of a system, pursuant to the inclusion criterion described in Section 2.2.2. Yet, many of these studies did not reference systems thinking, and exclusively relied on common qualitative data collection techniques such as interviews and focus group discussions or quantitative methods such as regression analyses of cross-sectional survey data (see for instance, [28,29,30,31,32]).

A decision was made to narrow the definition of systems thinking at the full-text review stage to reduce inter-coder conflict and enhance study replicability, which is a key consideration for scoping reviews. Based on existing reviews on the application of systems thinking in health contexts and generally, it was decided that studies would only be included for data extraction if they included at least one of the commonly applied systems thinking theories, methods, or tools in Figure 1 [2,4,33].

While Figure 1 does not contain an exhaustive list, it captures systems thinking approaches commonly adopted in the health discipline, as identified in existing reviews [2,4,33]. The implications of adopting the narrower definition are set out in the Discussion below; however, a list of all studies excluded at the full-text screening stage is provided in Appendix 1 for transparency, and to assist future researchers who are interested in a broader conceptualization of systems thinking. The fact that the definition of systems thinking adopted at the title and abstract screening stage was broader means that there is no risk that relevant studies were excluded at that stage.

### 2.4. Data Extraction and Analysis

Data were extracted from included papers using a data extraction tool comprising the following categories: author(s); year of publication; specific STI(s) studied; population characteristics (e.g., gender, sexuality, age, ethnicity, membership of other priority groups), and systems thinking approach adopted.

The task of extraction was shared between two reviewers (MR and DV) with cross-checking for consistency. The extracted data were organized in tabular and narrative form, as set out in the Results section. A critical appraisal of study quality was not conducted; a critical appraisal is not standard/required practice for scoping reviews [12,13].

## 3. Results

As shown in Figure 2, *n* = 6102 studies were yielded from the search of databases and supplementary sources. After removing duplicates, the titles and abstracts of *n* = 4732 studies and the full text of *n* = 433 studies were screened against inclusion/exclusion criteria. The number of included studies was *n* = 70. Study details are summarized in Table 1 and elaborated on below.

The majority of included studies were conducted in African nations (*n* = 34), particularly South Africa (*n* = 14) and Kenya (*n* = 13). The next most common setting for studies was North America (United States of America *n* = 19 and Canada *n* = 3), followed by Asian nations (*n* = 5), Latin American and Caribbean nations (*n* = 3), and Western European nations (*n* = 3). Only one Australian study was identified, and the remainder of the study settings were either global or unspecified.

Generally, studies examined STI prevention and control in the population at large; exceptions to this were studies concerned with people living with HIV (*n* = 22) [5,9,34,35,36,37,38,39,40,41,42,43,44,45,46,47,48,49,50,51,52,53], people who inject drugs (*n* = 5) [54,55,56,57,58], women (*n* = 16) [5,11,59,60,61,62,63,64,65,66,67,68,69,70,71,72], children and young people (*n* = 11) [43,49,57,72,73,74,75,76,77,78,79], members of LGBTQIA+ communities (*n* = 8) [34,42,43,79,80,81,82,83], African American communities (*n* = 3) [42,59,60], other communities of color (*n* = 2) [42,67], sex workers (*n* = 2) [11,66], people in custodial settings (*n* = 1) [59], and Indigenous people (*n* = 1) [66].

Ninety-three percent (*n* = 65) of studies were focused on HIV [5,9,11,34,35,36,37,38,39,40,41,42,43,44,45,46,47,48,49,50,51,52,53,54,55,56,57,58,59,60,61,62,63,64,65,66,67,69,70,71,74,75,76,77,78,79,81,82,83,84,85,86,87,88,89,90,91,92,93,94,95,96,97,98,99], and three studies had a general sexual health/STI focus [69,73,77]. Two studies focused specifically on chlamydia; Viana and colleagues examined the consequences that increased chlamydia screening and diagnosis could have on the capacity of a health service to adequately treat clients, while Teng and colleagues modelled population dynamics to identify an optimal age-specific chlamydia screening strategy [10,72]. The two syphilis studies included in this review examined ways of improving early syphilis diagnosis through the analysis of social networks [80], and behaviour-over-time analyses to examine training needs associated with the rollout of point-of-care testing [68].

Of the HIV-focused studies, 33 were concerned with treatment [5,35,36,37,38,40,43,44,45,46,47,48,49,50,51,52,53,56,57,58,59,60,67,69,70,71,74,84,87,91,92,93,98]. Of these, 17 examined aspects of the treatment cascade (e.g., commencement and/or adherence) [5,35,37,43,44,45,48,49,50,51,52,53,57,67,69,74,87], 12 investigated the concept of treatment as prevention including in the context of vertical transmission [36,38,52,56,57,58,59,60,70,71,84,91,98], four focused on factors contributing to HIV drug resistance [40,46,92,93], and one concerned the provision of care to terminally ill patients [47].

Twenty-one studies had an HIV testing focus [5,37,42,43,49,50,51,56,64,65,69,70,71,74,85,87,94,95,96,97,98], of which one was focused on home- or self-testing [85]. Twenty-five studies focused on issues related to HIV harm reduction, including intentional transmission [9], stigma [41], sexual behaviours and ‘high risk’ behaviours [38,39,58,59,61,76,77,78,79,89,90], female condom use [11], PrEP [39,42,62,81,82,83,98], injecting drug use [54,56,58], and general preventative measures [63,66].

Most studies were focused on high-level or abstract systems—namely, society-at-large or the health system in general terms. Seventeen studies applied systems thinking to specific health settings such as hospitals and health clinics [10,11,43,47,48,49,50,52,62,64,65,68,69,70,71,74,95]. Other settings included specific non-government organizations and community groups [42,52,75,78,86], prisons [59], schools [73,77], policy communities [45,51], sex on premises venues [11], and supervised drug injecting facilities [54].

The most commonly applied systems thinking method was system dynamics modelling, which was expressly mentioned in 28 studies [9,10,11,37,38,42,44,45,47,51,52,53,55,58,60,61,63,67,72,84,85,92,93,94,95,96,97,98]. Applications of the method included understanding how:
Policies and other interventions affect resources and/or public health outcomes [10,72,85,94,95,96,97];Socio-economic and environmental factors influence protective and/or risk behaviours [9,11,42,44,45,61,63];Individuals engage with and/or navigate health care systems [53,67];Local context and other implementation factors affect intervention outcomes [85].

Other systems thinking methods included network analysis, which was used in ten studies to understand program reach [75,82], or the transmission of HIV and other STIs within defined social and sexual networks [36,39,54,55,56,60,80,81]. Agent-based modelling was expressly mentioned in four studies as a method for understanding interactions between people who inject drugs and other individuals [56], incarcerated men and their female sexual partners [59,60], and young people living with HIV and HIV service providers [43]. Scenario technique/planning was used in two studies [76,94]. Burman and colleagues described how the scenario technique was used to build young peoples’ skills around managing sex and relationships in a high-HIV prevalence setting [76], while Kok and colleagues build scenarios into their modelling to assess how variations in budget and resource allocation impact testing practices and HIV epidemiology [94].

Fifteen studies incorporated causal loop diagrams [5,10,11,37,40,42,48,53,61,73,90,92,93,94,99] and five used stock and flow diagrams [10,38,67,96,97]. The diagrams were used to depict: effects of shortages in medicines and other supplies on health system functionality [5]; transmission pathways [10,37,38,42]; factors affecting the implementation of a health promotion intervention or effectiveness of service reforms [11,48,73]; factors affecting engagement in health help-seeking, transmission risks, and/or quality of life [53,61,90,96,97,99]; and factors affecting the emergence of drug resistance [91,92,93,94].

Concept maps were featured in five studies [48,49,56,66,86], and process maps were used in six studies [10,64,65,71,73,95]. The processes mapped were patient pathways through a clinic/service [10,64,65,71,95], and a sexuality education enhancement intervention [73]. One study incorporated a systemigram [35], and four related papers incorporated the Cynefin framework [76,87,88,89]. No studies used the root cause analysis technique.

**Table 1 ijerph-20-05708-t001:** Characteristics of included studies (*n* = 70).

Study Citation	Region (Country)	Population	STI or BBV Focus	Aspects of Prevention and Control	Level of System	Systems Thinking Tools and Approaches
Adams et al. [59]	USA	People in custodial settings People from ethnic groups (African American) Incarcerated men and female partners	HIV	Harm/risk reduction (sexual behaviours) Treatment (TasP)	General society/health system Other community setting (prisons)	Agent-based modelling
Adams et al. [60]	USA	People from ethnic groups (African American) Women	HIV	Harm/risk reduction Testing Treatment (TasP)	General society/health system	System dynamics modelling Agent-based modelling Network analysis Behaviour over time graphs
Adekola et al. [73]	Africa (South Africa)	Young people	Sexual health generally	Harm/risk reduction	Educational Institution (schools) Other (rural areas)	General systems theory Causal loop diagram Process mapping
Annadurai et al. [80]	Asia and Pacific (India)	LGBTIQA+ people (MSM)	Syphilis	Testing (early identification of syphilis)	General society/health system	Network analysis
Batchelder et al. [61]	USA	Women People living in impoverished urban communities	HIV	Harm/risk reduction (sexual behaviours)	Society at large	System dynamics modelling Causal loop diagram
Beima-Sofie et al. [74]	Africa (Kenya)	Young people	HIV	Testing Treatment (cascade)	Specific health service	Other (flow mapping)
Beltran et al. [81]	USA	LGBTIQA+ people (Gay and bisexual men)	HIV	Harm/risk reduction (PrEP uptake)	Other (US Military)	Network analysis
Bezemer et al. [34]	Western European (Netherlands)	People already diagnosed with an STI. LGBTIQA+ people (MSM)	HIV	Other (understanding transmission patterns)	General society/health system	Other (Bayesian time scale phylogenetic analysis)
Bond et al. [84]	Africa (Zambia, South Africa)	Population at large	HIV	Treatment (TasP)	General society/health system	System dynamics modelling
Boruett et al. [35]	Africa (Kenya)	People already diagnosed with an STI	HIV	Treatment (cascade)	General society/health system	Behaviour over time graphs Systemigrams
Bouchard et al. [54]	Canada	PWID	HIV Hepatitis B	Harm/risk reduction (injecting drug use)	Other (Supervised injecting facility)	Network analysis
Bousmah et al. [85]	Africa (South Africa)	Population at large	HIV	Testing (home-based HIV counselling and testing)	General society/health system	System dynamics modelling Behaviour over time graphs
Brown et al. [86]	Australia	Population at large	HIV	Other (peer-led programs with range of activities)	Community setting (Peer-led organisations)	Concept mapping
Burman et al. [75]	Africa (South Africa)	Young people Men	HIV	Other (negotiating HIV challenges)	Community setting (Waterberg Welfare Society)	Network analysis
Burman et al. [87]	Africa (South Africa)	Population at large Other (rural setting)	HIV	Testing Treatment (cascade)	General society/health system	Cynefin
Burman et al. [88]	Africa (South Africa)	Population at large Other (rural setting)	HIV	Other (reduce aggregate viral load in community)	General society/health system	Cynefin
Burman et al. [76]	Africa (South Africa)	Young people Other (marginalised community)	HIV	Harm/risk reduction (sexual behaviours)	General society/health system	Scenario technique/planning Cynefin
Burman [89]	Africa (South Africa)	Population at large	HIV	Harm/risk reduction (reducing impact of ABC legacy) Other (general study of HIV landscape)	General society/health system	Cynefin
Burman et al. [90]	Africa (South Africa)	Population at large	HIV	Harm/risk reduction (sexual behaviours)	General society/health system	Causal loop diagram Behaviour over time graphs Other (Dynamic prevention)
Caplon et al. [62]	USA	Women	HIV	Harm/risk reduction (PrEP)	General society/health system Specific health service (IPV, PrEP, and reproductive health providers	General systems theory
Czaja et al. [77]	USA	Young people Population at large	HIV Sexual health generally	Harm/risk reduction (sexual/risky behaviours)	General society/health system Global institution Community setting (range of settings including schools)	General systems theory
De Boer and Lutscher [63]	Other (not specified)	Population at large Male and female-specific models	HIV	Harm/risk reduction (general)	General society/health system	System dynamics modelling
du Toit and Craig [36]	General International	People already diagnosed with an STI	HIV	Treatment (TasP)	General society/health system	Network analysis
Dickson-Gomez et al. [37]	Latin American and Caribbean (El Salvador)	People already diagnosed with an STI Population at large	HIV	Testing Treatment (cascade)	General society/health system	System dynamics modelling Causal loop diagram
Eastment et al. [64]	Africa (Kenya)	Women	HIV	Testing	Specific health service (family planning clinics)	Process mapping
Eastment et al. [65]	Africa (Kenya)	Women	HIV	Testing	Specific health service (family planning clinics)	Process mapping Behaviour over time graphs
Eaton et al. [91]	Africa (South Africa)	Population at large	HIV	Treatment (TasP)	General society/health system	Behaviour over time graphs
Fujimoto et al. [82]	USA	LGBTIQA+ people (MSM) Men	HIV	Harm/risk reduction (PrEP)	General society/health system	Network analysis
Goncalves and Kamdem [38]	Africa (Cote d’Ivoire)	People already diagnosed with an STI Population at large	HIV	Harm/risk reduction (sexual behaviours) Treatment (TasP)	General society/health system	System dynamics modelling Stock and flow diagrams
Gupta et al. [39]	General international	People already diagnosed with an STI Population at large	HIV	Harm/risk reduction (PreP and sexual behaviours)	General society/health system	Network analysis
Harper et al. [78]	Africa (Kenya)	Young people	HIV	Harm/risk reduction (sexual behaviours)	Other (Rural community group)	General systems theory (Ecological Systems Theory)
Khan et al. [55]	USA	PWID	HIV	Other (how HIV is transmitted through population)	General society/health system	System dynamics modelling Network analysis Behaviour over time graphs
Kiekens et al. [92]	Africa (Sub-Saharan Africa, countries not specified)	Population at large	HIV	Treatment (drug resistance)	General society/health system	System dynamics modelling Causal loop diagram (feedback loops)
Kiekens et al. [93]	Africa (Sub-Saharan Africa, countries not specified)	Population at large	HIV	Treatment (drug resistance)	General society/health system	System dynamics modelling Causal loop diagram (feedback loops)
Kiekens et al. [40]	Africa (Tanzania)	People already diagnosed with an STI	HIV	Treatment (drug resistance)	General society/health system	Causal loop diagram
Kok et al. [94]	Canada	Population at large	HIV	Testing (optimal allocations of testing resources to minimise new infections)	General society/health system	System dynamics modelling Scenario technique/planning Causal loop diagram
Levy et al. [41]	Africa (Kenya)	People already diagnosed with an STI Population at large	HIV	Harm/risk reduction (stigma)	General society/health system	Behaviour over time graphs (simulation scenarios)
Long et al. [95]	Africa (Kenya)	Population at large	HIV	Testing	Specific health service (family planning clinics)	System dynamics modelling Process mapping
Lopez-Entrambasaguas et al. [66]	Latin American and Caribbean (Bolivia)	People from ethnic groups (Indigenous) Sex workers Women	HIV	Harm/risk reduction (general)	General society/health system	Concept mapping
Lounsbury et al. [67]	USA	People from ethnic groups (people of colour) Women	HIV	Treatment (cascade)	General society/health system	System dynamics modelling Stock and flow diagrams Behaviour over time graphs
Lutete et al. [42]	USA	People already diagnosed with an STI LGBTIQA+ people (gay, bisexual, and other MSM) People from ethnic groups (people of colour; African American)	HIV	Harm/risk reduction (PrEP) Testing	Community setting (social service providers)	System dynamics modelling Causal loop diagrams
Mabey et al. [68]	Africa (Tanzania, Uganda, Zambia) Asia and Pacific (China) Latin America and Caribbean (Peru, Brazil)	Pregnant women	Syphilis	Testing (point-of-care testing)	Specific health service (antenatal clinics)	Behaviour over time graphs
Marshall et al. [56]	USA	PWID Population at large	HIV	Harm/risk reduction (injecting drug use) Testing Treatment (TasP)	General society/health system	Agent-based modelling Network analysis Concept mapping Behaviour over time graphs
Martin et al. [96]	USA	Population at large	HIV	Testing (HIV testing laws)	General society/health system	System dynamics modelling Stock and flow diagrams Behaviour over time graphs
Martin et al. [97]	USA	Population at large	HIV	Testing (HIV testing laws)	General society/health system	System dynamics modelling Stock and flow diagrams Behaviour over time graphs
Mayhew et al. [69]	Africa (Kenya)	Pregnant women Population at large	HIV Sexual health	Testing Treatment (cascade)	Specific health services (integrated HIV and reproductive health service)	Behaviour over time graphs
McKay et al. [43]	USA	People already diagnosed with an STI Young people LGBTIQA+ people (MSM)	HIV	Testing Treatment (cascade)	Specific health service (linkage to care services)	Agent-based modelling
Omondi et al. [98]	Africa (Kenya)	Population at large	HIV	Harm/risk reduction (PrEP) Testing Treatment (TasP)	General society/health system	System dynamics modelling
Orievulu and Iwuji [44]	Africa (South Africa)	People already diagnosed with an STI	HIV	Treatment (cascade)	Other (rural community/climate)	System dynamics modelling
Orievulu et al. [45]	Africa (South Africa)	People already diagnosed with an STI	HIV	Treatment (cascade)	Policy/Community (government drought planning organisation)	System dynamics modelling
Palk et al. [83]	Western Europe (Denmark)	LGBTIQA+ people (MSM)	HIV	Harm/risk reduction (PrEP)	General society/health system	Behaviour over time graphs
Pedamallu et al. [9]	Other (unspecified)	People already diagnosed with an STI	HIV	Harm/risk reduction (disclosure of HIV-positive status)	General society/health system	System dynamics modelling
Riou et al. [46]	Africa (Botswana, Eswatini, Malawi, Mozambique, Namibia, South Africa, Zimbabwe, Lesotho)	People already diagnosed with an STI	HIV	Treatment (drug resistance)	General society/health system	Behaviour over time graphs
Rustagi et al. [70]	Africa (Côte d’Ivoire, Kenya, Mozambique)	Pregnant women	HIV	Harm/risk reduction (prevention of mother-to-child transmission) Testing (antenatal testing) Treatment (TasP)	Specific health service (health facilities in participating countries)	Behaviour over time graphs
Sherr et al. [71]	Africa (Kenya, Cote d’Ivoire and Mozambique)	Pregnant women	HIV	Testing (HIV testing for pregnant women and newborns) Treatment (TasP)	Specific health service (prevention of mother-to-child HIV transmission (services)	Process mapping Behaviour over time graphs
Stevens et al. [79]	USA	Young people People from ethnic groups LGBTIQA+ people (gay and transgender adults who identify as male at birth)	HIV	Harm/risk reduction (sexual behaviours)	General society/health system	Other (Ecological systems theory)
Sturmberg et al. [47]	Africa (Kenya)	People already diagnosed with an STI	HIV	Treatment (palliative)	Specific health service (Eastern Deanery AIDS Relief Program)	System dynamics modelling
Teng et al. [72]	USA	Young people Women	Chlamydia	Testing (screening)	General society/health system	System dynamics modelling
Topp and Chipukuma [48]	Africa (Zambia)	People already diagnosed with an STI	HIV	Treatment (cascade)	Specific health service (Primary health centres)	Causal loop diagram Concept mapping
Tuan [99]	Africa (South Africa)	Population at large	HIV	Other (general level factors leading to AIDS pandemic)	General society/health system	Causal loop diagram
Tulloch et al. [57]	Asia and Pacific (Thailand)	PWID Other (Children)	HIV	Treatment (cascade)	General society/health system	Other (System thinking lens using Yayo Bakoum framework)
Viana et al. [10]	Western Europe (UK)	Population at large	Chlamydia	Testing (screening) Treatment	General society/health system Specific health service (hospital outpatient clinic)	System dynamics modelling Causal loop diagram Stock and flow diagram Process mapping Behaviour over time graphs
Wagner et al. [49]	Africa (Kenya)	People already diagnosed with an STI Young people	HIV	Testing Treatment (cascade)	Specific health service (government health facilities)	Concept mapping
Wai et al. [50]	Canada	People already diagnosed with an STI	HIV	Testing Treatment (cascade)	Specific health service (Vancouver Coastal Health and Providence Health Care)	Other (Unified Modelling Language and State Machine Diagram)
Weeks et al. [11]	Asia and Pacific (China)	Sex workers Women	HIV	Harm/risk reduction (female condom use)	General society/health system Specific health service (local clinical and public health organisations) Other (sex work establishments)	System dynamics modelling Causal loop diagram
Weeks et al. [53]	USA	People already diagnosed with an STI Population at large	HIV	Testing (HIV test and treat continuum) Treatment (cascade)	General society/health system	System dynamics modelling Causal loop diagram
Weeks et al. [51]	USA	People already diagnosed with an STI	HIV	Treatment (cascade)	Policy community (Ryan White Planning Councils)	System dynamics modelling
Weeks et al. [52]	USA	People already diagnosed with an STI	HIV	Treatment (cascade; TasP)	Specific health service (HIV medical and social service providers) Community setting (HIV community organisations)	System dynamics modelling
Yaya Bocoum et al. [5]	Africa (Burkina Faso)	Pregnant women People already diagnosed with an STI Population at large	HIV	Testing Treatment (cascade)	General society/health system	Causal loop diagram
Zou et al. [58]	Asia and Pacific (China)	PWID	HIV	Harm/risk reduction (sexual behaviour; injecting drug use) Treatment (TasP)	General society/health system	System dynamics modelling

Notes: PLWHIV—People living with HIV. PWID—People who inject drugs. MSM—Men who have sex with men. ART—Antiretroviral treatment. TasP—Treatment as Prevention.

## 4. Discussion

This scoping review represents the first attempt to document how systems thinking has been applied in STI prevention and control literature. By specifically designing a search strategy focused on STIs, the review has been able to locate many studies not previously identified in existing reviews with a more general focus on public health [3,4]. For instance, compared to Carey et al. [3], who included articles published between 1990 and 2015, the broader nature of our search strategy enabled us to find more STI-specific articles than was the case in that review (*n* = 70 compared to *n* = 1). Regardless, extending the search strategy to include articles published prior to 2012 would have resulted in even more studies, and the temporal limitations of this review are acknowledged. Nevertheless, the results of this review suggest that targeted search strategies focused on specific diseases or health issues (such as STIs) can be useful in identifying examples of systems thinking in public health literature which are not captured by search strategies that only utilize more general keywords such as “health”, “public health”, and “population health”.

In making the case for more systems thinking reviews, it would be remiss not to caution researchers about the difficulties involved in developing definitions and inclusion criteria sensitive enough to capture systems thinking literature but specific enough to exclude general discussions of ‘systems’, as well as distinct but related concepts such as ‘complexity theory’ [100]. Peters notes that:

If there is a jungle of terminology used to describe scientific endeavor, it becomes even thicker in the area of systems thinking, perhaps because of its diverse heritage… It is based on a wide variety of scientific methods… It uses an even larger array of instruments or tools… The use of these terms is not consistent across or within scientific fields, including systems sciences, and the continuum from tool to method to theory and framework is often blurry.[2]

While this scoping review initially adopted Arnold and Wade’s systems thinking definition on the basis that it represented a synthesis of existing approaches, it proved difficult to apply to the task of systematic screening [24]. Consequently (as described in the Methods section), the protocol was revised to include a list of common theories, methods and tools that would be deemed to constitute systems thinking. However, the revised approach may have resulted in the omission of systems thinking studies that applied less common (but nevertheless relevant) tools, methods, theories, or terminologies, or even studies in which the authors did not appreciate that they were adopting systems thinking approaches. The approach may have also resulted in the inclusion of studies that applied some systems thinking methods or tools (e.g., behaviour over time graphs) but which may not be considered by their authors (or others) to represent an application of systems thinking.

While our results must be read in light of these limitations, they also highlight the urgent need for greater conceptual clarity around how systems thinking literature can be efficiently searched for and screened in the context of scoping and other reviews. A “know it when you see it approach” [101] is inconsistent with scoping and systematic review methodology which requires “adequate explication of a core notion… [to enhance] both reliability and systematicity of the research” [102]. The difficulties we experienced have been noted in systems thinking reviews in other subject areas [4,103]. For Rusoja and colleagues:

“the “fuzzy” boundary between [systems thinking] and other fields of focus like socio-ecological frameworks, network science, action research, participatory research, quality/implementation/improvement science, health services research, team science, and realist reviews, amongst others, reflects not only a key limitation of [their] review but also a challenge to those in health in trying to understand and apply these ideas”.[4] (p. 604)

Despite the limitations described, this review provides a useful snapshot of current research gaps and priorities with respect to the application of systems thinking in the context of STIs. With the exception of the studies conducted in African nations, there was a paucity of STI systems thinking literature relating to regions known to have the highest age-standardized STI incidence rates—namely, Latin America and the Caribbean, Central Europe, Eastern Europe and Central Asia, South Asia, and parts of the Middle East [104]. Only three included studies were conducted in Latin America and the Caribbean, five in Asia, and none in Eastern Europe or the Middle East. It is possible that the English-language focus of this scoping review contributed to the low number of studies identified from non-Anglophone countries, and it is recommended that future reviews are conducted to identify literature in other languages. A recent study on stakeholder perspectives of systems thinking in Southeast Asia found that some participants believed that “the nomenclature of systems thinking, and interrelated concepts were all in English and predominantly adopted a Western approach to implementation, which could have (in part) served as an impediment to wide-scale adoption” in that region [105]. Collating more examples in which systems thinking has been applied in non-Anglophone regions may assist in overcoming the perception that it is a Western construct with little application to other global contexts.

There was also a dearth of studies focused on sub-populations known to be at higher risk of STIs. For instance, there is evidence to suggest that Indigenous populations in North America, Latin America, and Australia have higher rates of STIs compared to the general population [106]; however, only one included study focused on STI prevention and control in an Indigenous context (namely, the Ayoreo people of Bolivia) [66]. Similarly, people from migrant backgrounds have been identified as bearing a disproportionate burden of HIV compared to native-born populations [107], yet none of the included studies explicitly examined STI prevention and control in the context of migrant communities. The relevance of systems thinking to health disparities research is well-recognized, and examples of its application exist outside of the STI sector [108,109,110,111,112]. Moreover, it has been observed that “Indigenous peoples worldwide have been using sophisticated approaches that have great synergy with systems thinking for millennia” [113]. There is a need for more STI research employing systems thinking to understand how structures and processes in society interact to affect the health and health help-seeking behaviours of marginalized groups.

More systems thinking research is also needed on STIs other than HIV. The small number of included studies focused on STIs such as syphilis and antimicrobial-resistant gonorrhoea stands in contrast to the body of literature, demonstrating the increasing global significance of these conditions [114,115,116]. The utility of extending systems thinking beyond HIV to include other STIs is particularly evident when one considers the epidemiologic synergy between them, and the way that innovations in one space (i.e., the introduction of PrEP) can create issues and opportunities for the prevention and control of other STIs [117,118].

Despite the emergence of key innovations since 2012, this review found that comparatively few studies applied systems thinking to understand issues relating to PrEP or HIV self-testing technologies. However, as Atun notes, systems thinking lends itself to nuanced analyses of the influence and uptake of innovations:

Multiple interacting factors influence the adoption of innovations, ranging from new technologies to novel service delivery models and to health policies. Therefore, a broader and more sophisticated analysis of the context, health system elements, institutions, adoption systems, problem perception, and the innovation characteristics within these will enable better understanding of the short- and long-term effects of an innovation when introduced into health systems. [119] (p. iv7).

Applying systems thinking to self-testing is particularly important in light of the World Health Organization’s recommendation for the technology’s use in efforts to close the gap in the number of undiagnosed people living with HIV [120]. Documented barriers to scale-up represent an interplay of political, economic, infrastructural, clinical, and individual factors, suggesting the utility of systems-level analyses of the issue [121].

The results of the review also point to the reliance on systems thinking methods and tools such as systems dynamics modelling, agent-based modelling, network analysis, causal loop diagrams, concept mapping, and process mapping. Other systems thinking methods and tools such as root cause analyses have been used effectively in other areas of public health and may warrant further exploration in an STI context [122,123,124].

Finally, since scoping reviews “do not aim to produce a critically appraised and synthesized result/answer to a particular question” [12], future researchers may wish to supplement the results of this study by conducting a systematic review. Such a review may assist in understanding whether there are any clinical or public health implications to the use of systems thinking approaches in the prevention and control of STIs. However, in developing a systematic review protocol, researchers should be mindful of the challenges involved in synthesizing studies utilizing a diverse range of methods across a number of countries/contexts and with regard to a number of STIs, and may wish to refine the scope of their inquiry accordingly.

## 5. Conclusions

While systems thinking has been employed in 70 studies relating to STI prevention and control since 2012, this review has identified large research gaps; in particular, systems thinking has been underutilized outside of African and North American contexts and has predominantly focused on HIV to the exclusion of other STIs of global public health significance. Scant attention has been given to new innovations in STI prevention and control including PrEP and the self-testing technologies, and there has been a tendency to overlook priority populations that bear a disproportionate burden of infections such as Indigenous populations and migrant communities. Through these findings, this review has set out a clear research agenda for the future application of systems thinking in the STI sector.

In addition, the review’s observations on the practical challenges of conducting a scoping review on the topic of systems thinking highlight the tensions inherent in applying systematic and replicable processes to investigate an amorphous concept that has evolved from a diverse disciplinary heritage to include a wide range of methods and tools. It is recommended that systems thinking subject experts and scoping/systematic review methodological experts collaborate to address the challenges identified so that the literature on systems thinking can be mapped efficiently, systematically, and accurately going forward. Consensus is needed on a definition of ‘systems thinking’ and a comprehensive list of systems thinking methods and tools is required to inform future search strategies. Authors and journal editors should also take care in ensuring that accurate and consistent keywords and subject headings are applied to systems thinking studies given the “fuzzy” [4] boundary that has been observed between systems thinking and other concepts including socioecological models, network science, and health services research.

## Figures and Tables

**Figure 1 ijerph-20-05708-f001:**
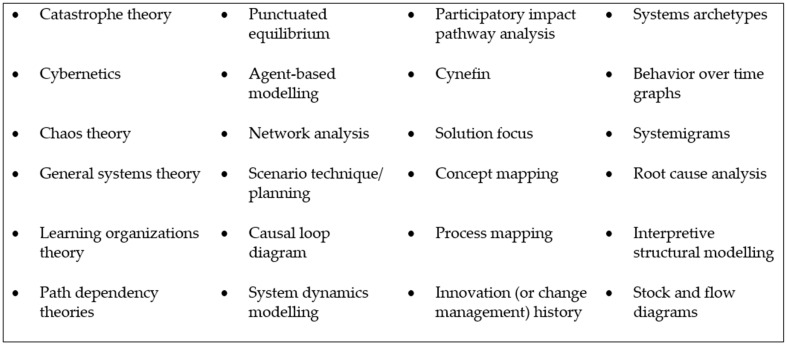
Systems thinking theories, methods, and tools.

**Figure 2 ijerph-20-05708-f002:**
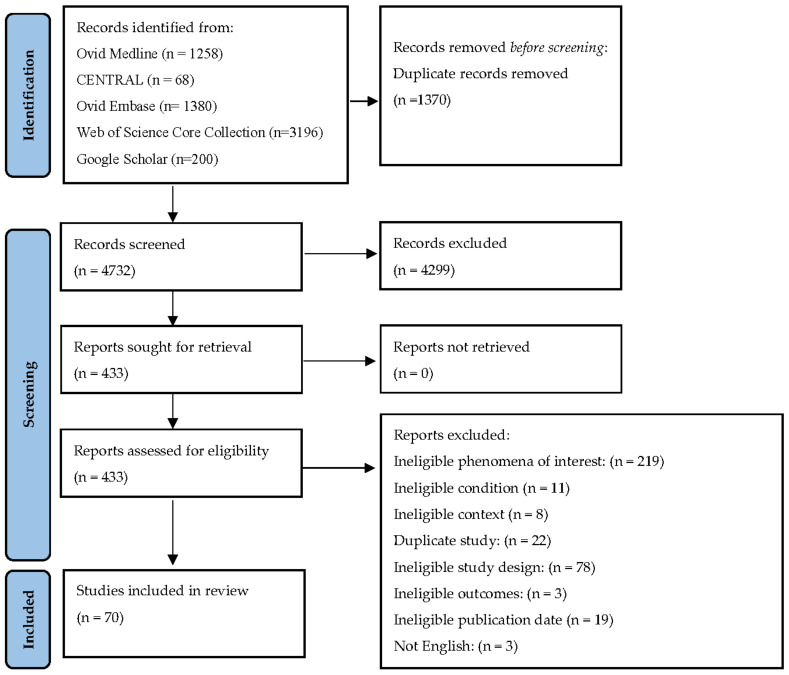
PRISMA flow diagram of search and study selection process.

## Data Availability

Data are available on request to the corresponding author.

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
