# Peer review of "The Application of Systems Thinking to the Prevention and Control of Sexually Transmissible Infections among Adolescents and Adults: A Scoping Review"

_ijerph, 2023, doi:10.3390/ijerph20095708_

Round 1

Reviewer 1 Report

The manuscript entitled “The application of systems thinking to the prevention and control of sexually transmissible infections among adolescents and adults: A scoping review” is a novel review aimed to identify papers published from 2012 in the area of prevention and control of STI using a “systems thinking” perspective, which is not commonly applied or clearly described in the literature. I think authors could add some lines, or examples and references related to the kind of main findings that  applying this methodology could allow compared to others.  Authors mention that themselves had to re-define the elements they had to consider in the inclusion criteria to better identify the manuscripts to review. I think this is a key point of the messages of the authors, “systems thinking” definition could be not enough delimited,  so adding a condition focused on the tools or methods typically used (Figure 1) was the strategy selected to choose the studies to full-text review. In this point, I think, authors could include references supporting the sentence “Based on existing reviews on the application of systems thinking in health contexts ..” from Section 3.3 lines 184-185, even if the figure 1 has listed some references related.

I think table 1 includes the topics defined in the specific research questions of the study, and I only will add in the discussion to analyze them could be a potential future study. A systematic review could be conducted using some of these selected articles,  to extract their main conclusions and the relationships found through the systems thinking, comparing the procedures, advantages and disadvantages of the methods used between the studies and in this way to contribute to a better understanding of this perspective. 

Author Response

Thank you for your considered and very helpful comments. Please see responses attached (particularly pages 2-3). We are hopeful that our amendments and explanations adequately address your concerns and suggestions. 

Reviewer 2 Report

The comments to the authors are attached.

Author Response

Thank you for your considered and very helpful comments. Please see responses attached (particularly pages 4-11). We are hopeful that our amendments and explanations adequately address your concerns and suggestions. 

Reviewer 3 Report

The application of systems thinking to the prevention and control of sexually transmissible infections among adolescents and adults: A scoping review

Thank you for the opportunity to review this paper which aims “to conduct a scoping review to identify literature in which systems thinking has been applied to the prevention and control of STIs among adolescent and adult populations globally. The purpose of the review is to answer the following research questions:

(1) Which specific STIs have been studied from a systems thinking perspective?

(2) In which populations were the studies conducted (i.e., country, age group, gender, sexuality, ethnic/Indigenous identity)?

(3) Which systems thinking concepts and/or tools have been utilized or referenced?”

General comment

This scoping review is clearly described with a good level of detail. It was a pleasure to read. The manuscript is publishable in its current form. I have only a few minor comments and questions.  

Title

Clear

Abstract

The abstract is clear and well-written.

Introduction

Clear and well-written.

The introduction says, “A search of Embase, the Cochrane Database of Systematic Reviews and JBI Evidence Synthesis suggests that no existing review on the subject exists or is under development.” Did you search review registers, such as PROSPERO?

Methods

It is a pleasure to read such a well-written and detailed methods section.

Results

The results are systematically and clearly presented.

Table 1

Clearly presented. Suggest ‘young people’ (describing a population group) is better than ‘youth’ (which describes a period).

Discussion

The discussion is very clear and fairly presented.

I wondered about including in the limitations, the omission of studies that were using systems thinking approaches that were not clearly identified as such by the authors.

Conclusion

Fair

Author Response

Thank you for your considered and very helpful comments. Please see responses attached (particularly pages 12-13). We are hopeful that our amendments and explanations adequately address your concerns and suggestions. 

Reviewer 4 Report

Summary

This scoping review investigated the application of systems thinking to the prevention and control of sexually transmissible infections among adolescents and adults. This study is interesting and can contribute to the current HIV prevention and care, but I still have some comments below.

Major Issues

Methods

1.     The authors only included literature since Jan 1 2012, using an argument of PrEP was introduced in US in 2012. However, I disagree with the notion of the authors, as this scoping review does not have a US focus. Therefore, the authors should not use a US timeline to conduct a scoping review. In addition, STI includes other infections beyond HIV, and there are other prevention methods available, too. I suggest the authors to extend their literature search and include more literature prior to 2012 to expand the temporal perspective in this scoping review.

Results

2.     In Table 1, I would use xxx et al, instead of the last name of the first author.

3.     In addition, it would be better for the readers to follow if the information of more concise study population each selected study included are included, such as adolescence vs. adult. I would suggest the authors to compare the results based on the study population of adolescence vs.adult, given the main focus (and title) indicated a population-level comparison.

Discussion

1.     Overall, I would suggest the author improve the text flow of the entire manuscript, but especially the discussion section, as there are a lot of long citing sentences the authors used as evidence for their statement. Yet, for me, and also other readers, this is new information and thus should fit better in the result section. The authors should use more concise language, and focus more on the potential public health/clinical implications of the findings from this review, instead of just repeating what has been found.

2.     The limitation section is clearly missing.

Minor Issues

1.     One formatting issue is that the authors used a lot of bullet points throughout the manuscript, which is not a comment in a public health journal. I suggest the authors to reformat these contents.

Author Response

Thank you for your considered and very helpful comments. Please see responses attached (particularly pages 14-20). We are hopeful that our amendments and explanations adequately address your concerns and suggestions. 
